# RNA-Seq Virus Fraction in Lake Baikal and Treated Wastewaters

**DOI:** 10.3390/ijms241512049

**Published:** 2023-07-27

**Authors:** Sergey Potapov, Anna Gorshkova, Andrey Krasnopeev, Galina Podlesnaya, Irina Tikhonova, Maria Suslova, Dmitry Kwon, Maxim Patrushev, Valentin Drucker, Olga Belykh

**Affiliations:** 1Limnological Institute, Siberian Branch of the Russian Academy of Sciences, Ulan-Batorskaya 3, 664033 Irkutsk, Russiabelykh@lin.irk.ru (O.B.); 2National Research Center Kurchatov Institute, Academician Kurchatov Square 1, 123098 Moscow, Russia

**Keywords:** transcriptomes, viruses, RNA, Lake Baikal, wastewater, bioinformatics, freshwater

## Abstract

In this study, we analyzed the transcriptomes of RNA and DNA viruses from the oligotrophic water of Lake Baikal and the effluent from wastewater treatment plants (WWTPs) discharged into the lake from the towns of Severobaikalsk and Slyudyanka located on the lake shores. Given the uniqueness and importance of Lake Baikal, the issues of biodiversity conservation and the monitoring of potential virological hazards to hydrobionts and humans are important. Wastewater treatment plants discharge treated effluent directly into the lake. In this context, the identification and monitoring of allochthonous microorganisms entering the lake play an important role. Using high-throughput sequencing methods, we found that dsDNA-containing viruses of the class Caudoviricetes were the most abundant in all samples, while Leviviricetes (ssRNA(+) viruses) dominated the treated water samples. RNA viruses of the families *Nodaviridae*, *Tombusviridae*, *Dicitroviridae*, *Picobirnaviridae*, *Botourmiaviridae*, *Marnaviridae*, *Solemoviridae*, and *Endornavirida* were found in the pelagic zone of three lake basins. Complete or nearly complete genomes of RNA viruses belonging to such families as *Dicistroviridae*, *Marnaviridae*, *Blumeviridae*, *Virgaviridae*, *Solspiviridae*, *Nodaviridae*, and *Fiersviridae* and the unassigned genus Chimpavirus, as well as unclassified picorna-like viruses, were identified. In general, the data of sanitary/microbiological and genetic analyses showed that WWTPs inadequately purify the discharged water, but, at the same time, we did not observe viruses pathogenic to humans in the pelagic zone of the lake.

## 1. Introduction

Aquatic microorganisms, such as bacteria, archaea, and viruses, are numerous and diverse components of freshwater and marine ecosystems. They are integral to nutrient cycling and energy turnover, making the study of their diversity, dynamics, and interactions critical to understanding all ecological aspects of aquatic ecology [1,2,3]. Viruses are considered significant but still poorly studied members of the microbial community.

Viruses are obligate intracellular parasites consisting of RNA or DNA molecules having various forms with or without the capsid protein; some of them have an additional lipid membrane shell. Viral communities in aquatic ecosystems are represented by viruses of eukaryotes, bacteria, and archaea; viruses of the latter two domains are commonly referred to as phages. Viruses are present both as free particles and insiders of the host cell in aquatic environments.

The first metagenomic approach to study DNA-containing viral communities in Lake Baikal was used in 2019 [4], and it included the study of a 0.2 µm fraction (approximately 30 m offshore at Bolshie Koty, southern basin). Taxonomic and functional diversity was later studied in a fraction below 0.2 µm from a pelagic layer of 0–50 m near the Listvyanka settlement in spring (ice cover) and summer [5]. The viral diversity was described in the epipelagic (photic), mesopelagic (aphotic), and bathypelagic (aphotic) zones in winter and summer. Significant changes in the composition of different viral communities between the epipelagic and bathypelagic zones of Lake Baikal have been reported [6]. The virome of the surface water near the Bolshye Koty settlement was found to be dominated by *Siphoviridae* and *Myoviridae* bacteriophages [7]. Later, the authors described and compared previously identified and new viromes from the pelagic zone collected during ice cover and three spring and summer periods. A cluster analysis indicated that viromes from Lake Baikal formed a cluster with viromes from the world’s largest (Michigan, Ontario, and Erie) and oldest (Biwa) freshwater lakes, which allowed for the identification of the World’s Largest Lakes (WLL) clade [8]. In 2021, a metagenomic analysis of DNA viruses on integral samples from four different deepwater and shallow stations evidenced a clear isolation of viral communities within different ecological zones [9]. The first transcriptome analysis of viruses in Lake Baikal revealed the predominance of transcripts belonging to DNA-containing bacteriophages, as well as representatives of a nucleocytoplasmic large DNA virus (NCLDV), including the closest relative of *Pithovirus sibericum.* Moreover, we identified a small number of sequences belonging to RNA viruses [10].

RNA viral communities are currently considered to be more diverse and complex than previously known, and their effect on community composition and global carbon flux into aquatic environments appears to be underestimated [11]. The initial evidence suggesting that RNA viruses may have significant ecological impacts was obtained by isolating viruses that infect various phytoplankton species, a basis for aquatic food webs. By now, researchers have identified RNA viruses that infect most of the basic photosynthetic Protist taxa. They include diatoms [12,13,14,15], dinoflagellates [16], *Raphidophyceae* [17], *Prasinophyceae* [18], and *Thraustochytriaceae* [19].

Autochthonous viruses are permanent inhabitants of pure oligotrophic waters, but their composition can change due to the transmission of allochthonous (alien) viruses by migratory birds; influx with river runoff; and anthropogenic pollution, especially from the influx of untreated sewage. One of the most important sources for the spread of viruses that pose a risk to human and animal health is birds. For example, by collecting cloacal swabs in the delta of the Selenga River, an important tributary of Lake Baikal, the authors identified several subtypes of avian influenza, as well as paramyxoviruses. The Baikal basin provides an important habitat for migratory birds from Northeast Asia, as three spring migration routes converge here, attracting millions of birds [20].

Viruses in wastewater have the potential to cause many diseases in humans and animals, for example, gastroenteritis, hepatitis, and some respiratory infections. The monitoring of water-transmitted viruses has a long history [21,22]. As previously reported, outbreaks of human diseases in aquatic environments are mainly caused by noroviruses (46%) and adenoviruses (24%) [23]. Thus, in the Great Lakes region, enteroviruses have been isolated from recreational beach substrates and ground waters for municipal use, indicating a higher risk to human health [24]. Waste and surface waters contaminated by sewage fluxes were found to contain high concentrations of adenoviruses and were not acceptable for recreation in the lower reaches of the Grand River (Michigan) [25]. Wastewater treatment plants are known to remove and inactivate pathogenic and conditionally pathogenic bacteria. Meanwhile, viruses such as noroviruses and rotaviruses are more resistant to processing than fecal bacteria [26].

Quantitative reverse transcriptase q(RT)-PCR is a routine procedure for the detection of human viruses (enteroviruses, noroviruses, etc.) in the environment, but it requires virus-targeted sequencing and is therefore complemented by transcriptomic analyses [27,28,29,30,31,32].

The study of wastewater samples using transcriptomics methods allowed for the monitoring of intestinal infections [33,34]. Moreover, this method has allowed for the identification of the composition of the viral community as a whole. For example, untreated wastewater samples collected in the United States, Spain, and Ethiopia contained 234 known viruses. The virome of the untreated wastewater was dominated by bacteriophages. Approximately 85% of the sequence reads were from 18 different species of the family *Virgaviridae*, and a number of representatives of the families *Dicistroviridae*, *Iridoviridae*, *Nodaviridae*, and *Parvoviridae* that infect insects were also identified. Three wastewater samples contained 17 viruses that can infect humans (*Adenoviridae*, *Astroviridae*, *Caliciviridae*, *Papillomaviridae*, *Parvoviridae*, *Picobirnaviridae*, *Picornaviridae*, and *Polyomaviridae*) [33]. A metagenomic analysis of viral communities from untreated wastewater in the USA (San Francisco, CA, USA), Nigeria (Maiduguri), Thailand (Bangkok), and Nepal (Kathmandu) revealed sequences belonging to 29 eukaryotic virus families infecting vertebrates, invertebrates, and humans [34]. In Kampala (Uganda), the same approach was applied to the study of wastewater and a swamp, detecting several human and vertebrate viruses, including Herpesvirales, *Iridoviridae*, *Poxviridae*, *Circoviridae*, *Parvoviridae*, and *Bunyaviridae* [35]. The virome from municipal untreated wastewater in San Adrian de Besos (Barcelona) had 41 viral communities, including pathogenic species of the families *Caliciviridae*, *Adenoviridae*, *Astroviridae*, *Picornaviridae*, *Polyomaviridae*, *Papillomaviridae*, and *Hepeviridae* [36]. F-specific (F+) RNA phages, a group of single-stranded RNA bacteriophages belonging to the family *Leviviridae*, are widely used as indicators of fecal contamination and/or intestinal viruses in water in several countries [37,38,39,40].

Lake Baikal is the largest oligotrophic water body, an enormous freshwater reserve (23,615 km^3^) [41], and a habitat for endemic hydrobionts [42]. In 1996, Lake Baikal was listed as a UNESCO World Heritage Site (https://whc.unesco.org/en/list/754/, accessed on 4 April 2023). It was found that the littoral zone of Lake Baikal and its shallow bays, currently exposed to growing human activities, suffer from extremely high fecal contamination from groundwaters [43].

Based on the above, we focused on performing a virus composition analysis on treated wastewater and Baikal water in an attempt to trace the route of viruses entering Lake Baikal with treated sewage water and to gain insights into the potential risks to human and hydrobiont health.

## 2. Results

### 2.1. Assessment of Microbial Quality of Effluent Wastewater and Water in Lake Baikal

An analysis of the treated water in all samples showed deviations from Russian sanitation regulations and hygienic standards [44]. In the town of Slyudyanka, the total coliforms (TCs) in the water discharged from the wastewater treatment plants in May exceeded the standards by four times. In August, microbial concentrations were all above the guide limits: total coliforms—1600 times, *Escherichia coli*—8000 times, enterococci—800 times, and coliphages—162 times. In August, the number of total coliforms in the treated water of the town of Severobaikalsk exceeded the standard limits by 1.8 times (Table 1). A sanitary/microbiological examination of the waters processed at the wastewater treatment plants of the towns of Slyudyanka and Severobaikalsk confirmed the ineffective disinfection of municipal wastewater.

The sanitary indicative microorganisms at the central sites of Listvyanka–Tankhoy, Ukhan–Tonkiy, and Elokhin–Davsha hydrological sections were consistent with the hygienic standards for surface waters. The studied bacteria were not detected in most of the samples and were present in the integral sample from the Ukhan–Tonkiy site (TC, *E. coli*, and enterococci—1 CFU/100 cm^3^) and from the surface layer at the Elokhin–Davsha site (TC—3 CFU/100 cm^3^); TC and *E. coli* was 1 CFU/100 cm^3^ at a depth of 5 m (TC and *E. coli*—1 CFU/100 cm^3^).

### 2.2. General Information

The number of reads after each processing step and the resulting contigs after assembly into SPAdes are shown in the Appendix A. The ORFs determined using the NR database were predominantly represented by bacterial transcripts (87.5–96.9%), and viral ORFs accounted for 0.6–5.1% (Figure 1). The proportion of unidentified sequences reached 25%.

The ratio of RNA and DNA viruses based on a Diamond analysis (IMG/VR) showed that DNA virus transcripts were the major contributors, except for in sample SRVP22_05 (Table 2).

### 2.3. DNA Viruses

#### 2.3.1. NR Database

The DNA viruses in all transcriptomes consisted mainly of double-stranded DNA bacteriophages of the families *Siphoviridae* (up to 89.5%), *Myoviridae* (up to 15%), and *Autographiviridae* (up to 14%). Sample SRVP22_05, which was dominated by the family *Myoviridae*, was the most contrasting. Single-stranded (ss) DNA viruses were represented by *Cruciviridae* (RVP4—1.7%, RVP5—0.5%, SevRVP22_08—0.7%, and SRVP22_05—2.8%), *Microviridae* (RVP4—1.3%, RVP6—2.2%, and SluRVP22_08—1.6%), and *Circoviridae* (RVP4—0.4%, RVP5—1%, SevRVP22_08—3.4%, SluRVP22_08—0.5%, and SRVP22_05—7%) families. The family *Geminiviridae* was detected only in RVP6 (0.5%), *Genomoviridae* and *Parvoviridae* only in SluRVP22_08 (0.5% each), and *Inoviridae* in SRVP22_05 (1.4%).

Cyanophage ORFs were solely found in pelagic samples represented in the largest amounts by the following taxa: *Synechoccus* phage S-SRP02 (isolated from a tropical freshwater lake), *Synechococcus* phage S-CBP2 (isolated from Chesapeake Bay water), *Synechococcus* phage S-SRP01 (Singapore Serangoon Reservoir), with the most similar proteins being photosystem II protein D2 (QPB08139)—94% amino acid (aa) identity, endonuclease (YP_009103176)—79.7% aa identity, and primase/helicase (YP_009103177)—79.4%. At the same time, hypothetical proteins comprised 45% of identifications with a 30% to 60.3% aa identity. All cyanophage ORFs belonged to the families *Autographiviridae* and *Kyanoviridae*, as well as to unclassified Caudoviricetes.

#### 2.3.2. IMG/VR Database

The ORFs were determined based on the contigs identified using VirSorter 2 and GeneMarkS. A comparison of the amino acid sequences against IMG/VR revealed the dominance of the dsDNA phages belonging to the class Caudoviricetes in all samples, with the *Autographiviridae* family being the second in number in RVP4 (Figure 2). Most Caudoviricetes sequences found in the IMG/VR database were classified to the class level only. Their number appeared to be higher than that in NR with similar Diamond settings; therefore, the choice of database was critical for the taxonomical identification. We used the contigs identified using VirSorter 2 to avoid false positives. Small amounts of Crassvirales representatives were found in the samples RVP4, RVP6, and SRVP22_05. CrAss-like phages are a diverse group of viruses that include some of the most abundant viruses of the human gut [45]. The presence of these phages in the pelagic samples was probably due to untreated sewage from vessels during the navigation season, a period of intense recreational activity. According to the State Small Vessels Inspectorate, approximately 300 heavy-lift vessels (http://geol.irk.ru/baikal/activ/mactiv2017, accessed on 4 April 2023) and 5.5 thousand small vessels are now in operation on Baikal, some of which discharge raw or untreated sewage into the lake.

A comparison of the results from RVP4, RVP5, and RVP6 obtained by searching IMG/VR revealed the total number of similar sequences (sequences from the Uncultivated Viral Genome (UVIG)) to be 106; SevRVP22_08, SluRVP22_08, and SRVP22_05—only 2; SluRVP22_08 and SevRVP22_08—142; and SRVP22_05 and SluRVP22_08—13 (Appendix A). As for the DNA virus composition, SevRVP22_08 and SluRVP22_08 shared 88 and 89 amino acid sequences, respectively, with sequences from pelagic samples, whereas SRVP22_05 had the lowest number of similar sequences: 4 with RVP4, 6 with RVP6, and 1 with RVP5. Judging by the DNA virus compositions, SRVP22_05 appeared to be the most divergent, whereas SevRVP22_08 and SluRVP22_08 showed the highest similarity. Most of the viral sequences of SRVP22_05 had hosts in the phylum Proteobacteria, and RVP4, RVP5, RVP6, SevRVP22_08, and SluRVP22_08 were dominated by phage sequences that infect Actinobacteria.

### 2.4. RNA Viruses

#### 2.4.1. NR Database

A total of 1901 ORFs belonging to the realm Riboviria were identified in all transcriptomes. The majority of ORFs were detected in the treated wastewater samples.

Positive-sense (ssRNA(+)) viruses appeared to be the most diverse class represented by 21 families in the following datasets (hosts according to literature data are shown in parentheses): *Dicistroviridae* (invertebrates), *Nodaviridae* (vertebrates and invertebrates), *Endornaviridae* (plants and fungi), *Virgaviridae* (plants), *Fiersviridae* (prokaryotes), *Tombusviridae* (plants), *Marnaviridae* (protozoa), *Astroviridae* (animals and humans), *Picornaviridae* (vertebrates), *Potyviridae* (plants), *Narnaviridae* (fungi), *Duinviridae* (prokaryotes), *Luteoviridae* (plants), *Solemoviridae* (plants), *Steitzviridae* (prokaryotes), *Blumeviridae* (prokaryotes), *Botourmiaviridae* (plants and fungi), *Iflaviridae* (insects), *Hepeviridae* (animals and humans), *Mitoviridae* (fungi), and *Betaflexiviridae* (plants and fungi).

Negative-strand (ssRNA(−)) viruses assigned to the family *Qinviridae*, which infect invertebrates, were found in the lowest amount. The amino acid sequences belonged to RdRp Fitzroy Crossing qinvirus 1 (QLJ83493) and the hypothetical protein Wuhan insect virus 15 (YP_009342457), the viruses of which were detected only in SRVP22_05.

The dsRNA viruses represented in the datasets included five families: *Picobirnaviridae* (animals and humans), *Totiviridae* (fungi and protozoa), *Partitiviridae* (plants, fungi, and protozoa), *Birnaviridae* (salmonid fish, birds, and insects), and *Cystoviridae* (bacteria), which were mainly present in the samples of SluRVP22_08 and SRVP22_05. In SluRVP22_08, *Picobirnaviridae* dominated (32% of all RNA viruses identified to the family level).

In addition, unclassified viruses, i.e., viruses identified only to the “virus” level or classified as Riboviria, comprised 18.5% to 100% of the samples.

The sample SRVP22_05 collected from the treated water at the wastewater treatment plant in the town of Slyudyanka in May was dominated by ORFs with sequences similar to those from activated sludge [46] (695 ORFs) and belonged to the class Leviviricetes. These sequences were identified as coat proteins (25.6–100% aa identity), *RdRp* (26.3–98% aa identity), and maturation proteins (23.2–98.4% aa identity). In addition, 265 ORFs were identified as *Leviviridae* sp.-related proteins, among which hypothetical proteins (141 proteins) with a 22.2–70.6% aa identity, 90 RdRp proteins (25–77.4% aa identity), 16 coat proteins (30.7–48.5% aa identity), and maturation proteins (32.9–39.8% aa identity) and replicase (29.1–49.6% aa identity) dominated. Thirty-three ORFs with a 24.7–61.4% aa identity belonged to the realm Riboviria, with no lower ranks. Fifteen proteins corresponded to the hypothetical proteins of Hubei levi-like virus 2 with a 35.1–73.6% similarity.

In 2020, the family Leviviridae was renamed Fiersviridae [47], but some sequences in the NR database remained Leviviridae. Therefore, we used the sequence names from the NR database.

The sample SluRVP22_08 was dominated by ORFs similar to those of ssRNA(+) phages Leviviricetes (102 ORFs, aa identity 23.9–100%), Marmot picobirnavirus (26 ORFs, aa identity 27.4–51.9%), and *Leviviridae* sp. (22 ORFs, aa identity 28.6–62.3%).

The sample from the Severobaikalsk wastewater (SevRVP22_08) was dominated by Picornavirales sp. (29.5–51.5% aa identity) and the *Trichosanthes kirilowii* picorna-like virus (33.8–42.2% aa identity) and defined to the realm Riboviria sp. (30.4–93.4% aa identity).

Viruses were found in negligible amounts in samples collected in the pelagic zone of Lake Baikal, in particular, the ORFs assigned to Hubei noda-like virus 4 (31.7–34.5% aa identity), Hubei tombus-like virus 37 (30–30.2% aa identity), Beihai noda-like virus 19 (28.7–53.9% aa identity), and some others in RVP4; the amino acid sequences of Beihai noda-like virus 19 (49.3–55.2% aa identity), Beihai noda-like virus 2 (26.6–42.9% aa identity), and Shahe picobirna-like virus 1 (27.7–27.8% aa identity) in RVP5; and the sequences of Beihai noda-like virus 2 (34% aa identity) in RVP6. It is intriguing that most of the sequences identified are similar to those detected in invertebrates from China [48].

#### 2.4.2. IMG/VR Database

The annotation of ORFs with IMG/VR yielded 861 sequences belonging to 23 RNA virus families, as well as 422 unclassified (to the family level) amino acid sequences. Sequences belonging to the family *Fiersviridae* were found predominantly in the treated wastewater samples SluRVP22_08 and SRVP22_05, but none was found in the pelagic samples (Figure 3).

The families *Nodaviridae*, *Tombusviridae*, *Dicitroviridae*, *Picobirnaviridae*, *Botourmiaviridae*, *Solemoviridae*, and *Endornaviridae*, and unclassified (to the family level) Riboviria, Orthornavirae, Kitrinoviricota, Durnavirales, Picornavirales, Ghabrivirales, and Nodamuvirales were detected in the RVP4 sample. The sequences in the RVP5 sample were represented by the families *Picobirnaviridae* and *Nodaviridae* and unclassified Riboviria, Orthornavirae, Durnavirales, Picornavirales, and Ghabrivirales. RVP6 had the smallest number of ORFs associated with RNA viruses and only three sequences represented by unclassified Durnavirales and *Marnaviridae*.

The families *Virgaviridae*, *Steitzviridae*, *Duinviridae*, *Solspiviridae*, *Partitiviridae*, and *Hepeviridae* were identified only in SluRVP22_08 and SRVP22_05. The largest number of ORFs corresponding to representatives of the order Picornavirales was found in the sample SevRVP22_08. Sequences belonging to the order Durnavirales (dsRNA infecting eukaryotes) were detected only in the pelagic samples.

In the pelagic samples, 63.9% of the sequences corresponded to ssRNA(+) viruses, and 36.1% of the identified RNA viruses corresponded to dsRNA viruses. In the treated wastewater samples, 97.5% of the ORFs were assigned to ssRNA(+) viruses and 2.5% to dsRNA viruses. It should be noted that the ssRNA(−) ORFs (family *Qinviridae*) detected according to the NR database were not identified with the IMG/VR database because ORF-containing contigs were not identified using VirSorter 2.

For the first time, we report the presence of detected viruses in the lake water despite the low number of RNA viral sequences from the pelagic zone of Lake Baikal.

### 2.5. RNA-Dependent RNA Polymerase Analysis

Open reading frames matching the database of intact genes encoding RNA-dependent RNA polymerase (RdRp, amino acid level) showed a wide range of similarity from 22.3% to 100%. In total, 672 sequences of RdRp were identified in this way from “The RNA Viruses in Metatranscriptomes” (RVMT) database (Appendix A), with lengths greater than 150 aa (RVP4—33, RVP5—12, RVP6—3, SRVP22_05—480, SevRVP22_08—33, and SluRVP22_08—111).

The maximum length of the RdRp sequence at the amino acid level was 2771 aa (RVP5). Overall, five phyla were identified: Lenarviricota, Kitrinoviricota, Pisuviricota, Duplornaviricota, and Negarnaviricota (Figure 4), with 72.7% of sample SRVP22_05 belonging to Lenarviricota.

The number of unique sequences in SRVP22_05 was 325, and the largest number of common taxonomic units was recorded between SRVP22_05 and SluRVP22_08 (45), i.e., in the treated wastewater samples collected in the town of Slyudyanka in May and August. Leviviricetes comprised 53% of the 45 common taxa.

In addition, the identified 672 RdRp sequences were annotated using the RefSeq database. A total of 554 *RdRp* were identified, with the majority of sequences (313) belonging to the family *Fiersviridae* and derived primarily from the samples SRVP22_05 (88.9%), SluRVP22_08 (10.8%), and SevRVP22_08 (0.3%). The treated wastewater samples were dominated by *Escherichia* virus FI (24.4–50.9% aa identity), *Escherichia* virus Qbeta (28.4–55.8% aa identity), and *Caulobacter* phage phiCb5 (24.7–44.5% aa identity) from the family *Fiersviridae*; 164 sequences were unassigned, and the closest relatives were identified as Hubei tombus-like virus 36 (30.8–45% aa identity), Changjiang tombus-like virus 22 (30–40.7% aa identity), and Changjiang picorna-like virus 11 (32.1–61.4% aa identity). In addition to bacteriophages (Leviviricetes), we also registered representatives of Pisuviricota, namely, Picobirnavirus dog/KNA/2015 (53.5–75.4% aa identity), Drosophila C virus (71.8–85% aa identity), and Human picobirnavirus (56–75.3% aa identity), and representatives of Kitrinoviricota, such as Tobacco mosaic virus (98–100% aa identity), Pepper mild mottle virus (PMMoV) (98–99% aa identity), Cucumber green mottle mosaic virus (97.9–100% aa identity), and Tomato brown rugose fruit virus (99.9–100% aa identity). PMMoV is a plant virus found in human feces, treated and untreated wastewaters, and aquatic environments contaminated by human feces [49]. Over the past decade, PMMoV has been proposed as a potential viral indicator of fecal pollution in marine and riverine waters [50,51].

The planktonic samples included such close relatives as Drosophila C virus (60.5% aa identity), Beihai picorna-like virus 107 (60.6% aa identity), and Hubei tombus-like virus 1 (46% aa identity).

In general, a set of intact *RdRp* sequences from the RVMT database showed more matches, as the sequences from the database (77,510 *RdRp*) outnumbered those from the RefSeq database (5767 *RdRp*).

In this work, a taxonomic tree based on the RdRp protein, the most representative RNA bacteriophage group (Leviviricetes), showed that the sequences obtained were distributed throughout the tree demonstrating wide diversity (Figure 5). It should be noted that we found no geographical confinement of the samples. For instance, the sequences from the NR database included representatives sampled elsewhere, namely, in Japan, China, the USA, and Austria, and this may attest to the cosmopolitan distribution of these viruses, testifying different sources of their retention, such as rice fields, soil, pond sediments, and activated sludge.

Only 135 of the 375 *RdRp* sequences obtained in this study, annotated as Leviviricetes in the RVMT database, were selected for tree construction because they overlapped the conserved region, which varied in length from 200 to 268 aa.

### 2.6. Search for the Complete Genomes of Viruses

VirSorter 2 was used to identify 3049 contigs in all samples and 1174 contigs in the SRVP22_05 sample. Twenty-six contigs were determined to be the most complete compared with their closest relatives using the blastn program, taking into account similarity, coverage, and aligned region length. Information on the closest relatives, coverage, and similarity can be found in Appendix A.

In view of the data on the closest relative, these sequences referred to such RNA virus families as *Dicistroviridae*, *Marnaviridae*, *Blumeviridae*, *Virgaviridae*, *Solspiviridae*, *Nodaviridae*, and *Fiersviridae*; an unassigned genus Chimpavirus; and unclassified picorna-like viruses. All whole or nearly whole genomes belonged to the realm Riboviria. The sequence length varied from 3035 nucleotides (nt) to 9467 nt. We also found a Boolarra virus segment RNA1 (AF329080), with a nucleotide identity of 99%. The contigs shown in Figure 6 presumably represent the complete genomes of the identified viruses. The highest similarities were found with Tomato brown rugose fruit virus isolate Tom1-Jo (KT383474)—99.9% nt identity, Tomato mosaic virus isolate Queensland (AF332868)—99.5% nt identity, Cucumber green mottle mosaic virus (D12505)—98.9% nt identity, and Pepper mild mottle virus (M81413)—98.2% nt identity. The trees based on deduced amino acid RdRp sequences from 26 genomes and GenBank sequences are given in Appendix A, generally showing high node support values.

### 2.7. Identification of Human and Animal Viruses

#### 2.7.1. Human Viruses

According to the Virus-Host DB database based on the ID taxa of the closest relatives (blastp, NR), the following human virus-related ORFs were detected in the wastewater collected from Slyudyanka in May (SRVP22_05): Circular ssDNA virus sp. (isolate HV-CV1, YP_009259556) with an aa identity of 31.7%, Human fecal virus Jorvi3 (YP_009389535) with an aa identity of 29.4–42.2%, Husavirus sp. (YP_009333306) with an aa identity of 26.5%, Human astrovirus (NP_059443) with an aa identity of 95.8–98.9%, and Human picobirnavirus (YP_239360, YP_239361) with an aa identity of 23.5–75.3%.

The sample collected in August from Slyudyanka (SluRVP22_08) contained the following: Salivirus FHB (YP_009067077) with an aa identity of 97.9–100%, Aichi virus 1 (NP_047200) with an aa identity of 96.9%, Human picobirnavirus (YP_239360, YP_239361) with an aa identity of 23.3–72.5%, Human papillomavirus 4 (NP_040890, NP_040891, NP_040893, NP_040894, NP_040895) with an aa identity of 99.3–100%.

A treated wastewater sample from the town of Severobaikalsk (SevRVP22_08, August)) contained Astrovirus VA4 (YP_006905856) with an aa identity of 26.3%.

No human-associated virus sequences were identified in the samples from the pe-lagic zone of Lake Baikal, based on the Virus-Host DB database.

#### 2.7.2. Animal Viruses

ORFs with a similarity to the proteins of viruses known to infect animals identified from the NR database generally had low resemblance to the known ones (under 50%, aa level). This probably indicates the lack of analogous sequences in the databases. Nonetheless, the highly similar amino acid sequences detected in this study give a first insight into their availability.

Viruses infecting the representatives of *Drosophilidae* (vinegar flies), *Otariidae* (fur seals), *Bovidae* (bovids), *Suidae* (swine), *Tephritidae* (fruit flies), *Aphididae* (aphids), Apidae (bees), *Thripidae* (thrips), and *Canidae* (canids) are shown in Table 3. A complete list of the identified ORFs of animal viruses is given in Appendix A.

### 2.8. Cluster Analysis of Transcriptomes

Clustering performed with the Metafast program showed that SRVP22_05 (Slyudyanka, May) was the most distant from the samples studied (Figure 7), consistent with the taxonomic analysis shown previously. The difference in the number of reads in the samples does not affect the analysis because the program provides the normalization of the data based on the total number of k-mers. The samples RVP5 and RVP6 from the pelagic zone of Lake Baikal proved to be the closest. The sample of treated wastewater from Severobaikalsk (SevRVP22_08) was part of a cluster with the pelagic samples. It is likely that clustering is caused by the fact that water for the Severobaikalsk central water supply comes from the nearby Tyya River, and, hence, municipal wastewater, as part of river water, contains allochthonous viruses similar in composition to those of the lake water. In addition, the treated wastewater from Severobaikalsk had a lower concentration of fecal bacteria; i.e., it was purified more effectively. In contrast, the water supplied to Slyudyanka is from underground water sources.

Clustering was verified using reads mapped with Bowtie 2 (human DNA removal) and SortMeRNA (ribosomal gene removal). The clustering characteristics remained unchanged.

The only sample grouped with other wastewater samples was SRVP22_05, which formed a cluster with the untreated wastewater collected in Southern California (the USA). This is probably because of the presence of RNA-containing viruses similar to those observed in these samples; for example, in a previous study, as well as in ours, tomato brown rugose fruit virus (to 66%), pepper mild mottle virus (10.6%), cucumber green mottle mosaic virus (10.4%), tomato mosaic virus (4.8%), and others were detected [31]. 

### 2.9. Functional Analyses of Viral and Non-Viral ORFs 

#### 2.9.1. VOG Database

According to the VOG database, the following amino acid sequences were detected in transcriptomes: RVP4—3193, RVP5—4007, RVP6—3962, SRVP22_05—3939, SevRVP22_08—3319, and SluRVP22_08—3522 (Appendix A). In all samples, the most abundant of the identified ORFs were the probable iron transport system ATP-binding protein HI0361 (VOG19828), exonuclease subunit 2 (VOG00052), and dTDP-glucose 4,6-dehydratase 2 (VOG00143).

The ORFs similar to those of the ATP-binding protein HI0361 (RVP4—219, RVP5—338, RVP6—327, SevRVP22_08—269, SluRVP22_08—259, and SRVP22_05—145) were compared to the amino acid sequences from the NCBI NR database showing similarities with ABC transporters (aa identity):In pelagic samplesBacterial domain—63.9–100% (median—100%), with various representatives of Actinobacteria, Proteobacteria, Firmicutes, Nitrospirae, and the eukaryote (99.6–100%) Malassezia restricta.In treated wastewaterBacterial domain—47.7–100% (median 99.8%), with Actinobacteria; Proteobacteria; Firmicutes; Bacteroidetes; Verrucomicrobia; Nitrospirae; and eukaryotes (33.1–100%), such as *Malassezia restricta*, *Symbiodinium microadriaticum*, and *Brachionus angularis*.

A comparison focusing on the virus database of the same sequences against GenBank revealed maximal similarity with the following: In pelagic samplesStaphylococcus phage PhiSepi-HH3, putative ABC transporter ATP-binding protein (QPB07827) with up to 99.4% aa identity; Klebsiella phage ST11-VIM1phi8.2, peptide transport system ATP-binding protein—up to 92.3%; Planktothrix phage PaV-LD, ABC transporter (ADZ31540)—up to 65.5% identity and others; similarity range from 24.9 to 99.4% (median 36.3%).In treated wastewaterStreptococcus phage MissG2, UvrABC system protein A (UJD17646) up to 71.1% aa identity; Escherichia phage vB_EcoS-640R1, lipoprotein-releasing system ATP-binding protein (URC10021) up to 70.3%; Klebsiella phage ST11-VIM1phi8.2 (QBP28525.1) up to 64.1%; and others.

The detected ABC transporters may be auxiliary metabolic genes (AMGs) and horizontal transfer genes. As previously suggested, bacteriophages mediate efficient gene transfer, including the transfer of genes possibly responsible for the mixotrophic lifestyle of *Tychonema* sp. [52].

Viral ORFs annotated using the VOG database include probable membrane antigen 75, the major capsid protein, the tail fiber assembly protein, the minor capsid protein P30, RNA-directed RNA polymerase, the coat protein, and packaging_enzyme_P4.

#### 2.9.2. KEGG Database

The KEGG pathway classification of the transcriptomes revealed that the following metabolic categories are the most abundant: “Carbohydrate metabolism”, “Protein families: genetic information processing”, “Protein families: signaling and cellular processes”, and “Translation” (Figure 8).

In the category “Carbohydrate metabolism”, the pelagic samples were dominated by the following: Glutamine synthetase (an essential enzyme in cellular nitrogen metabolism);Acetyl-CoA C-acetyltransferase (an enzyme that catalyzes the final step of fatty acid oxidation).

The transcripts in the effluent samples were dominated by the following: The 2-oxoglutarate dehydrogenase E1 component (involved in the tricarboxylic acid cycle);Acetolactate synthase I/II/III large subunit (a protein found in plants and microorganisms that catalyzes the first step in the synthesis of branched-chain amino acids).

In the category “Protein families: genetic information processing”, the following were predominant: RVP4—chromosome partitioning protein (required for efficient plasmid and chromosome partitioning in many bacterial species);RVP5, SevRVP22_08—DNA gyrase subunit A (belongs to the group of topoisomerases);RVP6—DNA segregation ATPase FtsK/SpoIIIE (mediates proper chromosome segregation in dividing bacteria);SRVP22_05—elongation factor G (prokaryotic elongation factor involved in protein translation);SluRVP22_08—DNA gyrase subunit B.

In the category “Protein families: signaling and cellular processes”, the most abundant proteins were as follows: In the RVP4, RVP5, RVP6, SevRVP22_08, and SluRVP22_08 samples—the ABC-2 type transport system ATP-binding protein;In SRVP22_05—OmpA-OmpF porin, the OOP family (the most abundant in the outer membranes of many Gram-negative bacteria).

In the category “Translation”, the following proteins were the most abundant:In the RVP4 and SevRVP22_08 samples—leucyl-tRNA synthetase;In RVP5, RVP6—isoleucyl-tRNA synthetase;In SRVP22_05—large subunit ribosomal protein L2;In SluRVP22_08—large subunit ribosomal protein L14.

In SRVP22_05 SRVP22_05, “Cell motility”, including RNA polymerase primary sigma factor, flagellar hook-associated protein 2, and chemotaxis protein MotB, and “Neurodegenerative disease”, with the dominant dynein axonemal heavy chain, were the most divergent categories.

In general, a functional analysis unveils the active metabolic patterns in bacterial life and their crucial role in the biological decomposition of organic substances.

## 3. Discussion

As expected, the number of bacterial transcripts was high (up to 96.9%). Despite the methodological complexity of separating virus particles or viral nucleic acid from bacterial nucleic acid, we were able to identify the viral sequences of both DNA and RNA viruses in the transcriptomes studied. Different approaches to virus identification were applied using the NCBI NR and IMG/VR databases. RNA viruses were identified according to the RVMT and RefSeq databases, comparing ORFs and *RdRp.* In our opinion, this method enhances the chances of achieving a more precise identification of viral sequences. At present, the databases are rapidly expanding, and their selection is of great importance for data interpretation, since the taxon is identified by performing a comparison with the already known sequences.

Here, we compared viral communities from effluent wastewaters discharged into Lake Baikal and viromes from its pelagic zone and searched for complete genomes. The samples collected were dominated by DNA viruses belonging to the class Caudoviricetes (dsDNA bacteriophages), mainly to the families *Siphoviridae*, *Autographiviridae*, and *Myoviridae*. Caudoviricetes are the most abundant and widespread viruses in natural ecosystems [53,54].

Untreated discharge in different localities (Pennsylvania, Barcelona, and Addis Ababa) was also dominated by DNA bacteriophages belonging to the families *Microviridae*, *Siphoviridae*, *Myoviridae*, *Podoviridae*, and *Inoviridae*. Approximately 85% of the sequence reads classified as identified viruses belonged to RNA viruses, namely, 18 different species of the family *Virgaviridae* [33]. Leviviricetes (ssRNA(+)) bacteriophages, along with Caudoviricetes, dominated the effluent flow running from the River Conwy catchment area (North Wales, UK) [28].

The effluent wastewater samples from the town of Slyudyanka located on the shore of Lake Baikal (SRVP22_05 and SluRVP22_08) contained RNA viruses (ssRNA(+)) similar to plant viruses (Tobamovirus), especially those infecting vegetables: tomatoes, pepper, and cucumbers. The obtained data are consistent with those in reports on the diversity and largescale distribution range of plant viruses in wastewater, the possible sources of which reside in agricultural discharge or human feces [31,33].

Here, we report an extremely diverse composition of ssRNA(+) viruses. Negative-strand viruses (ssRNA(−)) were present in negligible amounts, and dsRNA viruses accounted for 2.5% and 36.1% in treated wastewater and pelagic zone water, respectively. The samples SRVP22_05 and SluRVP22_08 were dominated by sequences belonging to the class Leviviricetes, apparently originating from a microbial community of activated sludge, since their closest relatives registered in GenBank are its inhabitants. Equal proportions of ORFs detected in the sample SevRVP22_08 belonged to Pisuviricota, Kitrinoviricota, and, to a lesser extent, Lenarviricota.

For the pelagic samples, the search in the IMG/VR database revealed *Nodaviridae*, *Tombusviridae*, *Dicitroviridae*, *Picobirnaviridae*, *Botourmiaviridae*, *Marnaviridae*, *Solemoviridae*, and *Endornaviridae*. Regretfully, it is generally a challenge to detect the host due to the absence of cultivated viral sequences in the databases. For instance, representatives of *Nodaviridae* are known to infect insects and fish in freshwater ecosystems [55] and cause the white tail disease of freshwater prawn [56]. The families *Dicitroviridae*, *Picobirnaviridae*, *Marnaviridae*, and *Nodaviridae* were encountered in freshwater Lake Tai (China) during an outbreak of a *Microcystis* spp. bloom, and RNA viruses comprised 42.5% of the total number of virus transcripts [57]. In Antarctica, Lake Limnopolar was dominated by the order Caudovirales (dsDNA viruses) and ssRNA(+) viruses of the family *Dicistroviridae* and the genus *Bacillarnavirus* (the order Picornavirales), and *Secoviridae*, *Marnaviridae*, *Iflaviridae*, *Potyviridae*, and *Tombusviridae* were found in smaller amounts [58]. Representatives of *Marnaviridae* are acknowledged pathogens of marine diatoms [14]. Until recently, our knowledge of RNA viruses in freshwater environments, especially in ancient waters, was strongly limited.

A taxonomic analysis showed that the viromes in the effluent wastewater samples SRVP22_05 and SluRVP22_08 were significantly different from other viromes based on the content of RNA viruses. In contrast, Baikal viromes (RVP4, RVP5, and RVP6) and treated wastewater collected in August in the town of Severobaikalsk (SevRVP22_08) and Slyudyanka (SluRVP22_08) demonstrated a close similarity based on the composition of DNA viruses.

Of the 3049 contigs identified using VirSorter 2, only 26 were putative complete genomes (including a segment similar to the Boolarra virus segment RNA1), with RNA viruses being their closest relatives, including terminal untranslated regions (UTRs), *RdRp* genes, capsid proteins, and maturation proteins. The maximum similarity was observed with the viruses belonging to the families *Dicistroviridae*, *Marnaviridae*, *Blumeviridae*, *Virgaviridae*, *Solspiviridae*, *Nodaviridae*, and *Fiersviridae*; the unassigned genus Chimpavirus; and unclassified picorna-like viruses. All genomes detected belonged to ssRNA(+) viruses 3 Kb to 9.5 Kb long infecting bacteria, plants, invertebrates, and, presumably, protozoans. One genome was not classified to the taxon level.

To compare transcriptomes as a whole, we used the Metafast program to cluster them. The samples from the pelagic zone of the lake were positioned separately on the tree. A previous UPGMA analysis of transcriptomes from Lake Baikal, based on a comparison of taxa from various natural sources (lakes, sea, and bays), supported the same isolated position of the Baikal samples in the dendrogram [10]. Bearing in mind the unique origin of the lake, the composition of hydrobionts, and hydrochemical and hydrophysical characteristics, the pronounced divergency of most Baikal viruses at the genetic level compared to the representatives of other aquatic ecosystems seems highly likely. The most distinct SRVP22_05 sample shared a clade with influent wastewater samples from Southern California (SB, OC, SJ, JWPCP, PL, NC, and HTP) [31].

In the effluent wastewater, we found several human-infecting viruses, and among them were enteroviruses (Aichi virus 1, Human astrovirus, and Astrovirus VA4) and viruses infecting the skin and mucosal epithelium (Human papillomavirus 4). Additionally, we detected the following human viruses, the ability of which to affect and induce diseases remains poorly studied: Human picobirnavirus, Husavirus sp., Human fecal virus Jorvi3, Salivirus FHB, and Circular ssDNA virus sp.

Aichi virus 1 (ssRNA(+), the genus Kobuvirus, the family *Picornaviridae*) is a human gastroenteritis agent transmitted via the fecal–oral route with contaminated food or water. This worldwide-spread virus is detected in various media: wastewater, river water, groundwater, and mollusks. The virus is found with a higher frequency and in larger numbers than any of the human enteroviruses. Aichi virus 1 may serve as an appropriate gastroenterovirus indicator [34,59,60].

Human astrovirus and Astrovirus VA4 (ssRNA(+), *Astroviridae*). The fact that astroviruses are pathogenic to humans was widely acknowledged by researchers in Thailand in 1991 [61]. Abundant evidence proved that astroviruses were a major cause of severe acute gastroenteritis in children, elderly people, and immunocompromised individuals (persons with a weakened immune system) [62,63,64].

Human picobirnavirus (dsRNA, *Picobirnaviridae*) was extracted from the stool of an infant with acute non-bacterial gastroenteritis in Thailand [65]. Picobirnaviruses have been found in different animal species, including invertebrates, and environmental samples. Since picobirnaviruses are ubiquitous in the feces/intestinal contents of humans and other animals with or without diarrhea, they are considered opportunistic pathogens of mammals and avian species, but the actual host remains unknown so far [66].

Husavirus sp. (ssRNA(+), *Picornavirales*) has been found in human feces and identified globally in different samples. Knowledge of the epidemiological and molecular features is hitherto fragmentary. Despite this limited understanding, all Husavirus sequences described have been detected in the stools of humans with different clinical manifestations: patients with a HIV-1-positive status, trachoma, acute diarrhea, and clinical silence. It is not yet known whether these viruses directly affect humans or other organisms in the human body; for instance, the hypothesis that helminths are natural Husavirus hosts was not supported [67,68].

Salivirus FHB (ssRNA(+), *Picornaviridae*) was present in the feces of 3.5% of ill children with diarrhea and 2.8% of clinically silent control patients. Saliviruses were observed from June to September during the warmest days and never on cooler days. No direct relationship between saliviruses and gastroenteritis has been reported. All known viruses may be found in the feces of children with or without gastroenteritis; however, the authors admit that gastroenteritis is caused by saliviruses at high viral loads [69]. Saliviruses were found in wastewater [34].

Human fecal virus Jorvi3 (ssDNA, the family *Circoviridae*) has been detected in human feces, yet there is no evidence that virus availability is associated with human illness. The authors suggest that infection with the virus may be beneficial to the host by preventing the development of autoimmune diabetes, but the studies need to be confirmed due to small sample sizes [70].

Human papillomavirus 4 (HPV) (dsDNA, the genus Gammapapillomavirus, *Papillomaviridae*) is typically found on the skin and oral mucosa of primates (https://ictv.global/report/chapter/papillomaviridae, accessed on 25 April 2023). HPV viruses are known to be involved in human cancer pathogenesis [71].

Circular ssDNA virus sp. has been isolated from pericardial fluid, but the authors found no association between this virus and pericarditis. It is suggested that the above-mentioned viruses may be replicated in human cells, presumably as opportunistic pathogens [72], notwithstanding that some circular replication-encoding single-stranded DNA viruses (CRESS-DNA) are animal pathogens and that high amounts of their representatives have been detected in samples of sick humans [72].

Coronavirus sequences were not detected in our dataset as in previous studies of wastewater [28]. At the same time, the use of respiratory-virus-enriched library preparation and sequencing has allowed for improvements in the detection of influenza A and coronaviruses [31]. Coronaviruses are able to survive in wastewater for hours or days while remaining contagious. The survival of coronaviruses in wastewater depends on many factors, such as temperature, pH, and treatment procedures. Wastewater treatment processes may inactivate or remove viruses, but viral RNA may still be detected in treated wastewater after a long time [73].

Human pathogenic viruses have not been identified at the central stations at Lake Baikal. Apparently, the concentration of fecal viruses decreases significantly with distance from the WWTP discharge point due to the dilution of the treated effluent flow by the lake water. As previously noted, viruses may spread over a large space from the discharge point [74]. However, when some of the virus particles reach a water body via wastewater, they degrade, disperse, and accumulate in hydrobionts, for instance, in mollusks [28]. Further observations of virome composition along the transect of sites at different distances from the effluent wastewater source points would allow for an evaluation of the variations in the composition and virus distribution pattern of viruses throughout the lake.

The microbiological examination of water on special seed culture media and a transcriptome analysis revealed the inadequate operation of WWTPs in the towns located on the shore of Lake Baikal. As previously reported, the concentrations of microorganisms serving as sanitary indicators were high in the coastal zone of the studied localities (the towns of Slyudyanka and Severobaikalsk) [75,76]. The sanitary indicators, including coliphages, registered at the pelagic sites of the lake complied with SanPiN requirements (the standards and sanitary rules of Russia), according to which the water from the surface sources should be free of enteroviruses.

Summing up, despite the inadequate treatment of wastewater discharged into Lake Baikal from the municipal WWTPs located on the lake shores, the present microbiological and virological parameters of the water in the deep pelagic zone meet the regulatory compliance requirements and pose no threat to human health.

Promoting recreation in the various bays of Lake Baikal demands further analysis that implies the use of complex methods, including microbiology, metagenomics, and metatranscriptomics approaches.

## 4. Materials and Methods

### 4.1. Sample Collection

Water samples of 100 L were collected from three pelagic sites of Lake Baikal in July and August 2022 at depths from 1 to 15 m (integral, 25 L from 1, 5, 10, 15 m depths). The sampling sites were located in three lake basins: RVP4—central station “Listvyanka settlement–Tankhoy settlement” (51°41.187 N, 105°00.096 E), RVP5—central station “Ukhan Cape–Tonky Cape” (52°53.117 N, 107°30.745 E), and RVP6—central station “Elokhin Cape–Davsha settlement” (54°25.531 N, 109°01.431 E). Treated wastewater in the volume of 5 L was sampled from the outlet treatment plant pipes in the town of Slyudyanka in May and August 2022, and in the town of Severobaikalsk in August 2022.

### 4.2. Sample Preparation

To obtain the virome, the samples were sequentially filtered through 0.4 and 0.2 µm polycarbonate filters (Sartorius, Göttingen, Germany; Reatrack-Filter, Obninsk, Russia) to remove detritus and zoo-, phyto-, and bacterio-plankton. The filtrates from each sample were concentrated with a tangential flow filtration VivaFlow 200 (Sartorius, Göttingen, Germany) to a volume of 100 mL, and then centrifuge concentrators (50 kDa) were used to further concentrate them to 1 mL, at 4 °C and 3000 rpm using VivaSpin Turbo 15 (Sartorius, Göttingen, Germany). The concentrate was frozen in liquid nitrogen and stored at −70 °C until further analysis.

Total RNA was isolated using ExtractRNA (Evrogen, Moscow, Russia) according to the manufacturer’s protocol. To prepare the RNA-seq library according to the MGIEasy RNAseq Library Prep Set protocol (MGI Tech, Shenzhen, China), 100–200 ng of isolated RNA was used. The following steps were performed: RNA fragmentation, reverse transcription, second chain synthesis, the polishing of dsDNA fragment ends, and adapter ligation (containing 10 nucleotide single-end indexes). Sequencing was run on the DNBSEQ-400 platform (MGI Tech, Shenzhen, China) with paired-end reads (2 × 150 bp).

### 4.3. Bioinformatic Analyses

The quality of Fastq files was visualized using the program FastQC v. 0.11.9 [77]. To remove human DNA sequences, the reads were mapped to a known genome, GRCh38_noalt_as (https://genome-idx.s3.amazonaws.com/bt/GRCh38_noalt_as.zip, accessed on 17 May 2023), using Bowtie 2 [78]. The program SortMeRNA (v. 4.3.6) [79] was used to remove the ribosomal RNA-encoding gene sequences of eukaryotes, bacteria, and archaea. The reads were assembled with SPAdes (rnaviral) v. 3.15.0 [80].

VirSorter 2 was used to identify the contigs belonging to RNA and DNA viruses [81]; only contigs above 500 nt were taken for analysis, as was performed in [82].

The open reading frames were identified using GeneMarkS (v. 3.36) [83].

The local version blastn 2.12.0+ (e-value 10^−5^) and the databases NCBI NT (release 250) and RefSeq (release v. 214) were used for the taxonomic annotation of the contigs. Diamond (v. 2.0.15) was used to annotate the ORFs [84], with the parameters *-more-sensitive*, *-min-score 50, -e-value 10^−5^*, using the database NCBI NR (release 250). The amino acid sequences obtained after processing in GeneMarkS from contigs after assembly in SPAdes were taken for analysis. Meanwhile, the taxonomic assignment was based on IMG/VR v.4 using Diamond with the parameters *-more-sensitive, -min-score 50*, *-e-value 10^−5^*; the amino acid sequences obtained after processing in VirSorter 2 and GeneMarkS were taken for analysis.

Protein functional annotations were generated using HHMER 3.2.1 (http://hmmer.org/, accessed on 17 May 2023) with the Virus Orthologous Groups Database (VOGDB), v. 213 (https://vogdb.org/, accessed on 30 January 2023), with the threshold parameter e-value 10^−6^ and the on-line service GhostKOALA (genus_prokaryotes + family_eukaryotes + viruses database; https://www.kegg.jp/ghostkoala/, accessed on 30 January 2023).

Heatmaps with dendrograms were drawn in the R programming language using the packages vegan v. 2.5-7 [85], gplots v. 3.1.3 [86], and viridis v. 0.6.2 [87] with normalization (method “total”); the visualization of the intersections was implemented using the package UpSetR v.1.4.0 (distinct mode) [88].

The search for *RdRp* genes in transcriptomes (in obtained ORFs, amino acids) was performed using the component local blastp 2.12.0+ (e-value 10^−5^) using the database of intact *RdRp* genes RVMT [89]. Sequences longer than 150 amino acids were used; the minimum threshold was chosen based on the database of intact *RdRp* genes, corresponding to the nearest minimum length of the sequence found in the database. The identified *RdRp* gene sequences were used for a phylogenetic analysis; alignment with sequences from NR, RefSeq, and RVMT was performed in the program MAFFT v. 7.407 [90], using the algorithm E-INS-i. The aligned sequences were manually checked, and the trimming of the beginning and the end of the alignment was performed using the program Mega 7 [91]. The tree was created using the program MrBayes (v. 3.2) [92].

RdRp protein coding sequences from the genomes of RNA viruses were aligned via MAFFT (v. 7.407), and the algorithm E-INS-i. TrimAl v. 1.2 (-gappyout) [93] and MEGA 7 were used to remove ambiguous regions. Phylogenetic trees were computed with IQ-TREE software v. 2.2.2.6 [94], model selection was performed using ModelFinder (http://www.iqtree.org/ModelFinder/, accessed on 17 May 2023) [95], and branch supports were obtained with the approximate likelihood ratio test (1000 repetitions) [96] and ultrafast bootstrap (1000 repetitions) [97]. The resulting trees were visualized and edited in iTOL (https://itol.embl.de/, accessed on 17 May 2023) [98].

A cluster analysis of transcriptomes with heat maps was performed in Metafast (v. 1.3.0) [99] with default parameters (SortMeRNA-processed reads were used as an input file). We then performed a comparative analysis based on the available transcriptome data [28,29,30,31] following the above described procedure.

Genome maps were generated with EasyFig v. 2.2.2, tBLASTx (e-value 10^−5^) applied for alignment [100].

Virus-Host DB [101] and VHost-Classifier [102] were applied for host identification.

### 4.4. Sanitary/Microbiological Analysis

The quality of the water processed at the wastewater treatment plants in Slyudyanka and Severobaikalsk was assessed according to the sanitary rules and norms of the Russian Federation [44,103]. The microbiological indicators were TC, *Escherichia coli*, enterococci, and coliphages; their limits were 500 colony forming units (CFU)/100 cm^3^, 100 CFU/100 cm^3^, 100 CFU/100 cm^3^, and 100 plaque-forming units (PFU)/100 cm^3^, respectively. The analysis and assessment of the water quality in Lake Baikal were carried out according to the hygienic standards [44,104]. We determined the number of TCs (≥500 CFU/100 cm^3^), *E. coli* (≥100 CFU/100 cm^3^), enterococci (≥10 CFU/100 cm^3^), and coliphages (≥10 PFU/100 cm^3^).

## 5. Conclusions

In this study, we analyzed DNA and RNA viruses from samples collected at three central stations on Lake Baikal and effluent wastewater discharge points. No human pathogenic viruses were registered in the samples from the pelagic zone, whereas the wastewater contained viruses known to infect humans. The treated wastewater samples contained plant viruses that may serve as complementary indicators of fecal contamination. In general, the RNA-seq method is suggested as an alternative approach for the detection of hazardous viruses.

## Figures and Tables

**Figure 1 ijms-24-12049-f001:**
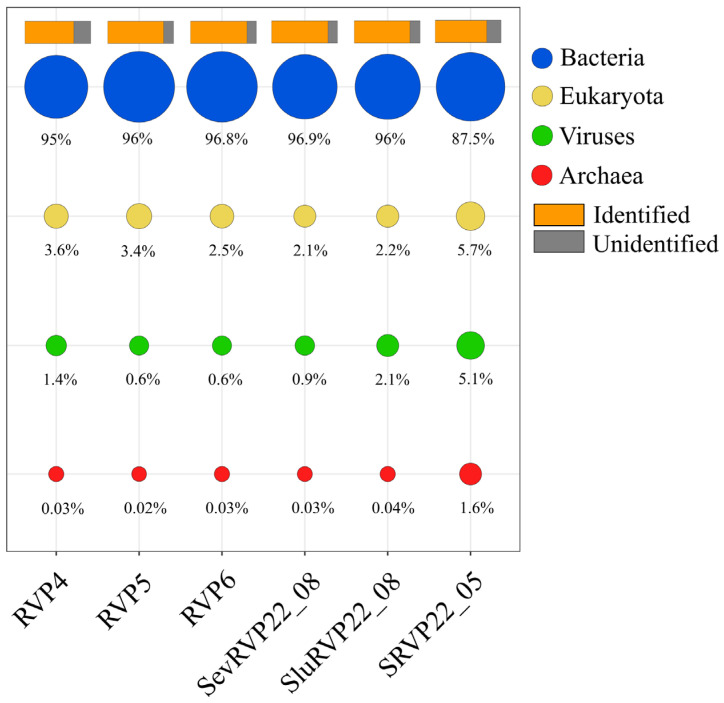
Percentage of ORFs identified in transcriptomes (NCBI NR database, blastp, e-value 10^−5^).

**Figure 2 ijms-24-12049-f002:**
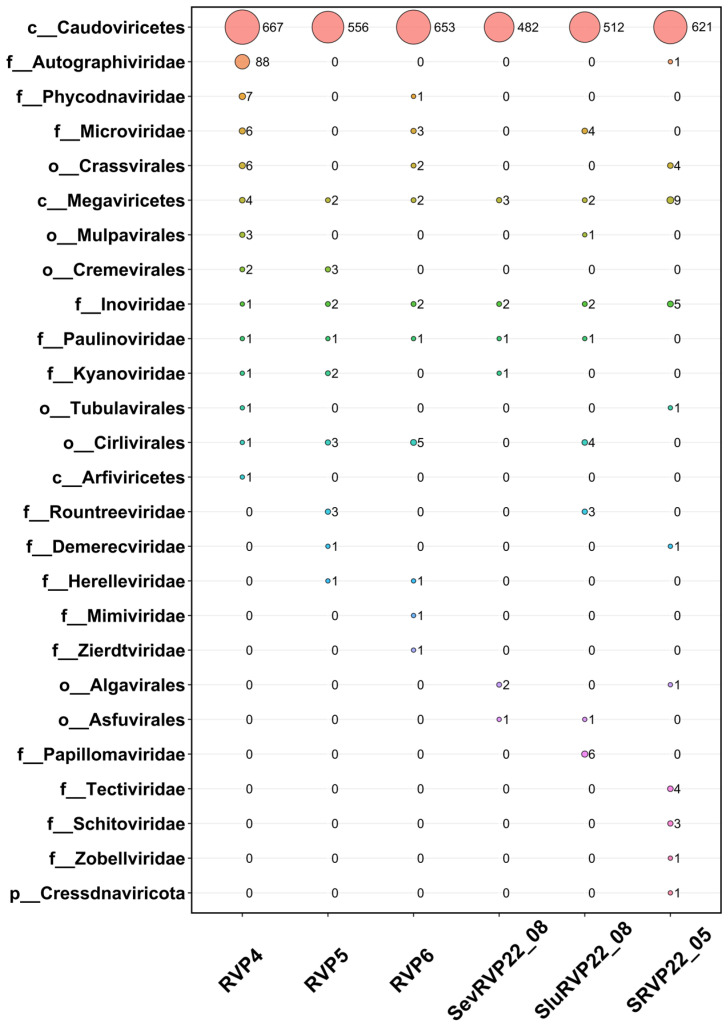
Bubble plot built upon IMG/VR database comparison of DNA viruses classified to the following ranks: f—family, o—order, c—class, p—phylum. Each color of the circle represents a taxon on the left.

**Figure 3 ijms-24-12049-f003:**
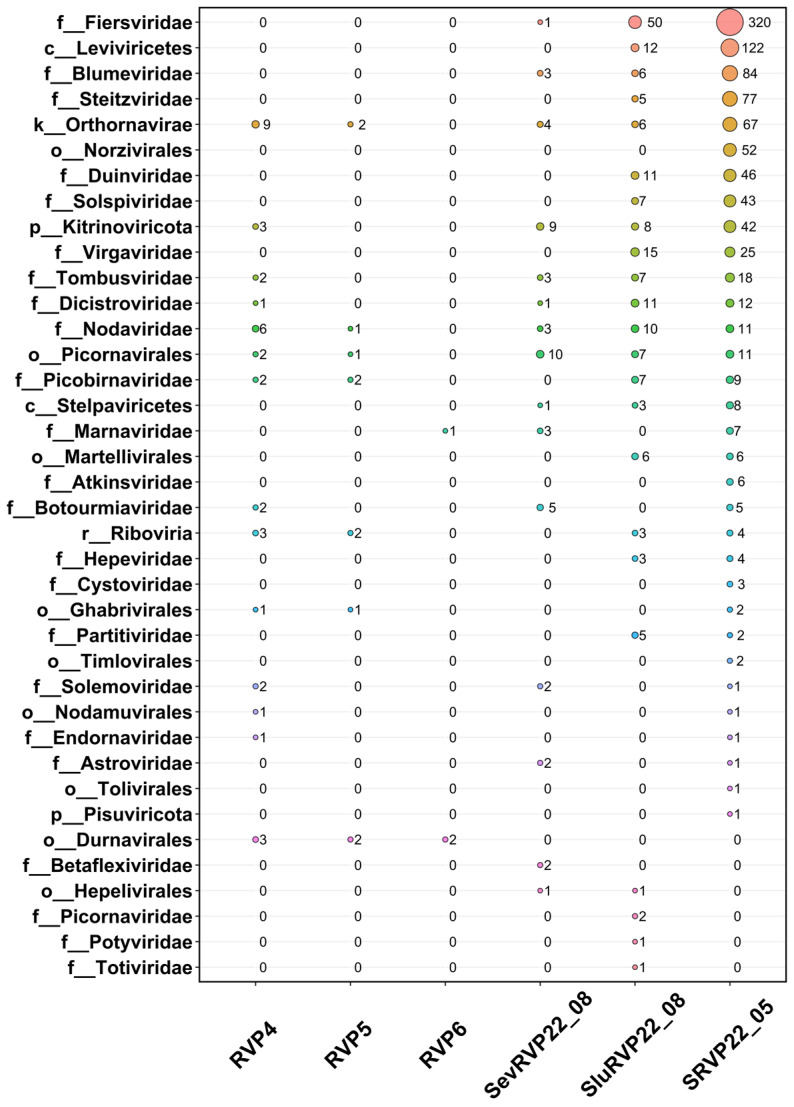
Bubble plot built via comparison with the IMG/VR database at the level of families and ranks of RNA-containing viruses to which the sequence was classified. f—family, o—order, c—class, p—phylum, k—kingdom, r—realm. Each color of the circle represents a taxon on the left.

**Figure 4 ijms-24-12049-f004:**
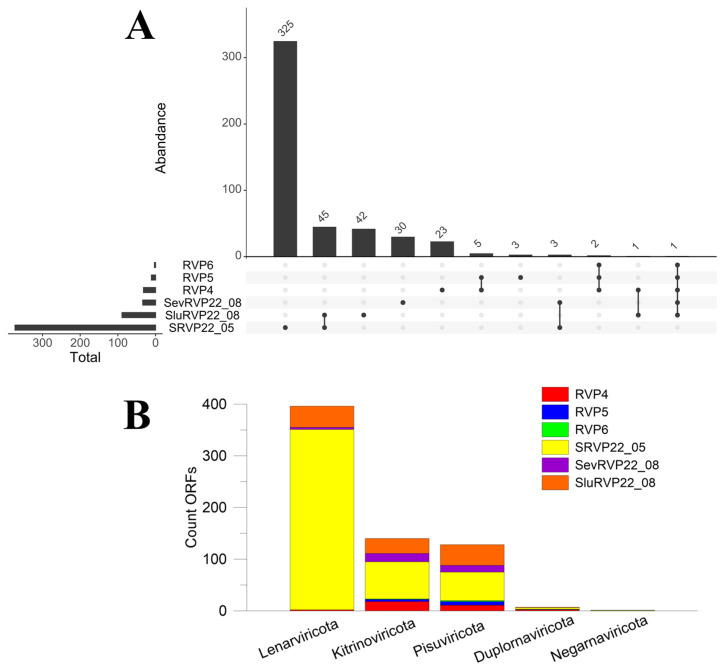
UpSet plot (**A**) based on the ORFs identified from the RVMT database: dots show unique ORFs present only in this sample, lines show common, Total—total number of unique ORFs identified in the sample (in contrast to the dots are ORFs that may be present in another sample); (**B**) bar plot based on identified ORFs (phylum level).

**Figure 5 ijms-24-12049-f005:**
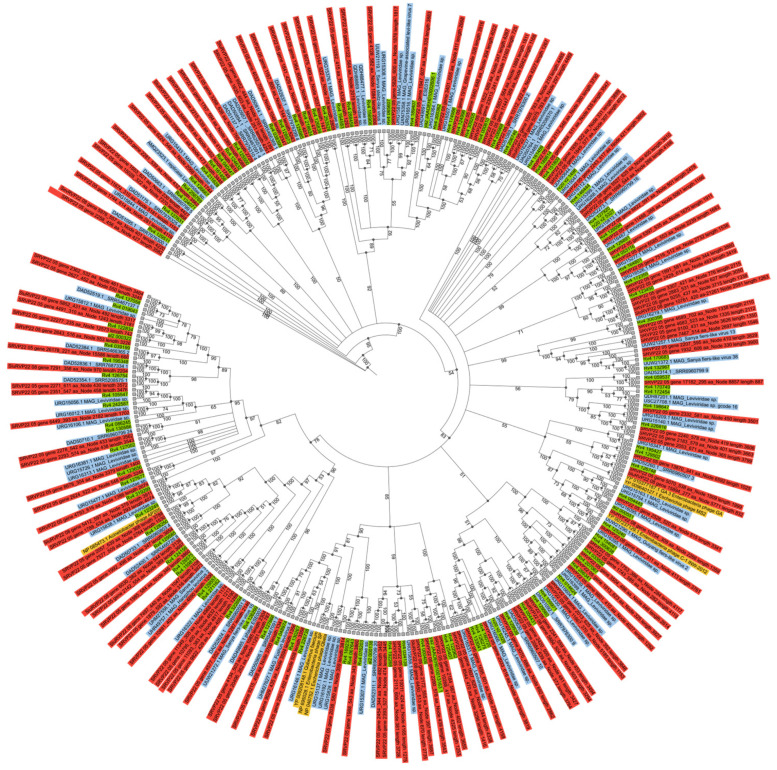
Taxonomic tree based on 342 *RdRp* Lenarviricetes sequences. Red represents this study, green—closest relatives from the RVMT database, blue—closest relatives from the NR database, yellow—closest relatives from the RefSeq database. Mean standard deviation: 0.03, potential scale reduction factor (PSRF+) = 1.0. Bootstrap values greater than 50% are shown at branches.

**Figure 6 ijms-24-12049-f006:**
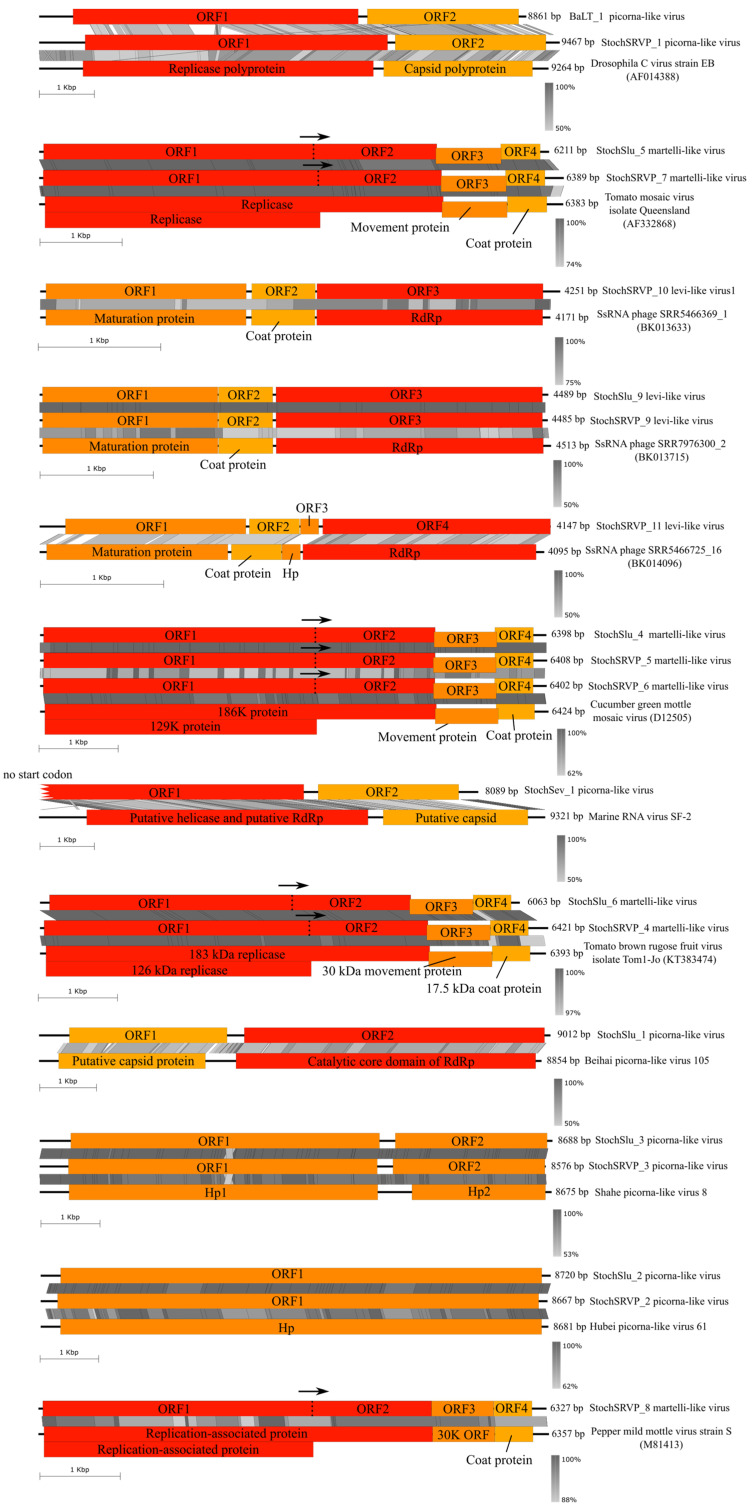
Maps of the putative complete genomes from this study and a comparison with the closest relatives (over 70% reference sequence coverage). Red indicates regions with *RdRp*, dark yellow—coat and capsid proteins, and orange—hypothetical proteins. Hp is a hypothetical protein. The arrow indicates expression via readthrough at stop codon. The right scale shows the percentage of sequence similarity (tblastx), and the left shows length in Kbp.

**Figure 7 ijms-24-12049-f007:**
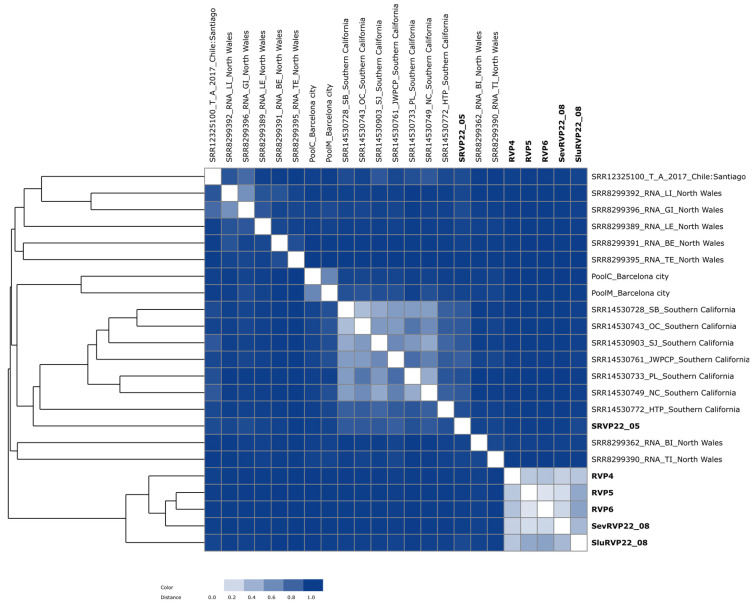
Heatmap and dendrograms based on transcriptome reads. Chile: Santiago [29]—raw sewage, North Wales [28]—influent (BI, LI, TI, GI) and effluent (BE, LE, TE) wastewater, Southern California [31]—influent wastewater (unenriched samples), Barcelona city [30]—raw sewage. Samples from this study are highlighted in bold.

**Figure 8 ijms-24-12049-f008:**
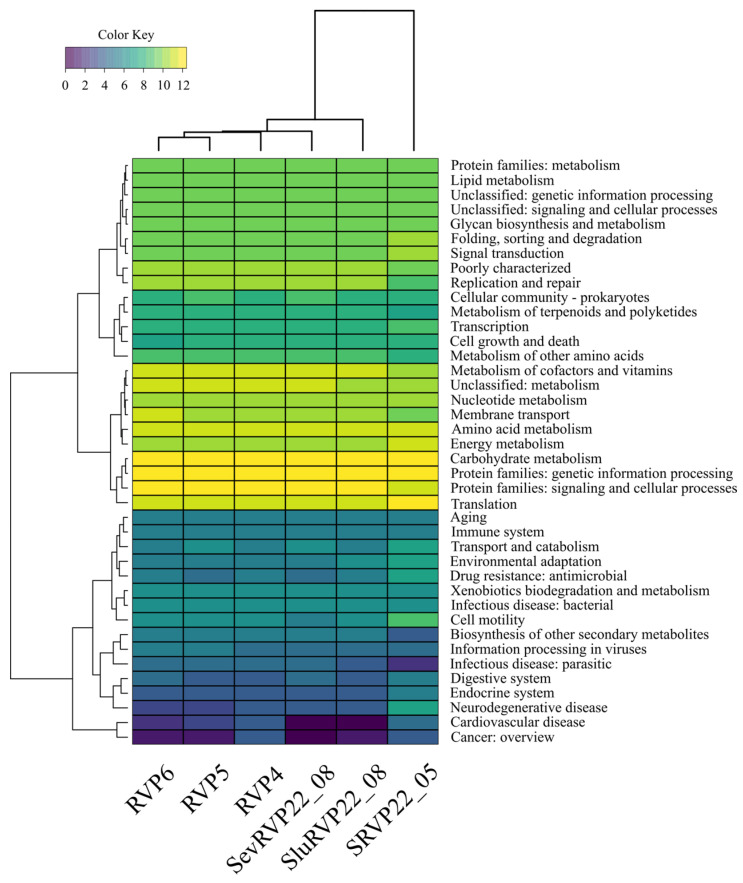
The most abundant categories identified in transcriptomes from Baikal and effluent wastewater according to KEGG pathway. Log10 scale.

**Table 1 ijms-24-12049-t001:** Number of sanitary indicative microorganisms in treated wastewater of the towns of Slyudyanka and Severobaikalsk.

Sample	TC, CFU/100 cm^3^	*E. coli*, CFU/100 cm^3^	Enterococci, CFU/100 cm^3^	Coliphages, PFU/100 cm^3^
Slyudyanka, May 2022	2000	0	24	0
Slyudyanka, August 2022	800,000	800,000	80,000	16,200
Severobaikalsk, August 2022	900	0	16	0

**Table 2 ijms-24-12049-t002:** Percentage of detected RNA and DNA viral proteins (IMG/VR database).

Sample	DNA Viruses, %	RNA Viruses, %
RVP4	95.4	4.6
RVP5	98.1	1.9
RVP6	99.5	0.5
SevRVP22_08	90.8	9.2
SluRVP22_08	74.2	25.8
SRVP22_05	39.6	60.4

**Table 3 ijms-24-12049-t003:** Putative hosts identified using Virus-Host DB (similarity over 60%).

Sample	Virus	Protein	Amino Acid Identity, %	Host
RVP4	*Dicistroviridae* sp.	QJI52080, capsid polyprotein	84.4	Drosophilidae
SRVP22_05	Drosophila C virus	NP_044946, capsid polyprotein	74.5–87.2	Drosophilidae
	Otarine picobirnavirus	AMP18960, RdRp	59.8–81.3	Otariidae
	Bovine picobirnavirus	ATY68940, RdRp	78.1–87.9	Bovidae
	Porcine picobirnavirus	ASM93467, RdRp	69.1–82.9	Suidae
SluRVP22_08	Drosophila C virus	QEQ50987, replicase polyprotein	74.5–89.3	Drosophilidae
	Bactrocera dorsalis picorna-like virus	QMU95558, putative polyprotein	55.4–74.6	Tephritidae
	Big Sioux River virus	ATI98941, structural protein precursor	83	Aphididae
	Apis mellifera associated microvirus 2	AZL82703, major capsid protein	74.9	Apidae
	Soybean thrips picorna-like virus 9	QQP18733, polyprotein	79.5	Thripidae
	Fox picobirnavirus	AGK45545, RdRp	56.6–64.7	Canidae
	Otarine picobirnavirus	AMP18960, RdRp	77	Otariidae
	Bovine picobirnavirus	AYF57589, RdRp	70.2–76	Bovidae
	*Picobirnavirus* sp.	QQM99847, putative capsid	29.1–66.7	Cervidae
	Porcine picobirnavirus	ASM93458, capsid protein	68.4–91.9	Suidae
	Phylloscopus inornatus ambidensovirus	QVW56839, MAG: putative structural protein VP1	65.7	Phylloscopidae

## Data Availability

Raw data obtained in fastq format are stored in the Sequence Read Archive (SRA) under the numbers PRJNA903674 (Lake Baikal) and PRJNA932453 (treated wastewater). Partial and complete genomes are stored in GenBank under the number OQ722313-OQ722338. Fasta files with *RdRp* gene sequences and sequences obtained after processing in VirSorter 2 can be found at https://github.com/SergeyBaikal/Characteristics-of-the-RNA-seq-virus-fraction-in-Lake-Baikal-and-treated-wastewater.git, accessed on 25 June 2023.

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
