# Peer review of "RNA-Seq Virus Fraction in Lake Baikal and Treated Wastewaters"

_ijms, 2023, doi:10.3390/ijms241512049_

Round 1

Reviewer 1 Report

The manuscript entitled “RNA-seq virus fraction in Baikal and treated wastewaters” by Potapov Sergey et al., performs a comprehensive analysis of transcriptomes of RNA and DNA viruses present in oligotrophic Lake Baikal water and effluent from wastewater treatment plants. As well as a description of the detected viruses.

In my view, the manuscript presents an excellent structure and development, with an adequate description of materials and methods. In addition, the manuscript collects and analyzes interesting information to increase the knowledge of the viruses most abundant in the lake, determining that the wastewater samples included viruses infecting humans and plant viruses, that may serve complementary indicators of fecal contamination. So, it would be very useful for researchers in areas such as virology, biotechnology, biological control of wastewater, basic science, biochemical, and for the scientific community in general. Therefore, it could be Accept in present form.

Just one suggestion:

- In figure 1, I think it would be better to change the color of Viruses and Archaea (Green / Red), so that they are not confused with the Identified / Unidentified indications (Green / Red).

Author Response

Dear Reviewer

We like to thank you for the review of our manuscript ijms-2497987

Reviewer

Response

The manuscript entitled “RNA-seq virus fraction in Baikal and treated wastewaters” by Potapov Sergey et al., performs a comprehensive analysis of transcriptomes of RNA and DNA viruses present in oligotrophic Lake Baikal water and effluent from wastewater treatment plants. As well as a description of the detected viruses.

In my view, the manuscript presents an excellent structure and development, with an adequate description of materials and methods. In addition, the manuscript collects and analyzes interesting information to increase the knowledge of the viruses most abundant in the lake, determining that the wastewater samples included viruses infecting humans and plant viruses, that may serve complementary indicators of fecal contamination. So, it would be very useful for researchers in areas such as virology, biotechnology, biological control of wastewater, basic science, biochemical, and for the scientific community in general. Therefore, it could be Accept in present form.

We thank the reviewer for such a high assessment of our work and will try to further expand our knowledge in such an interesting and significant subject.

In figure 1, I think it would be better to change the color of Viruses and Archaea (Green / Red), so that they are not confused with the Identified / Unidentified indications (Green / Red).

We thank the reviewer for this suggestion and  we've corrected the colors.

Reviewer 2 Report

                Over the last 4 years, there have been numerous papers published regarding the viruses present in Lake Baikal using NGS approaches.  The current study focuses on wastewater discharged into the lake.  Interestingly, the wastewater analysis revealed ineffective disinfection based on coliform counts in some towns.  Not surprisingly, most of the transcriptomes identified had their basis in dsDNA bacteriophage.

                Overall, this is a straightforward, descriptive study of viral nucleic acids in these wastewater samples.  It should serve as foundational information for specialists interested in the topic.  I do have a few suggestions to polish the presentation:

Minor Points:

1.        The manuscript should be very carefully reviewed for optimal use of the English language.

2.       The introduction, while it contains a great deal of important information regarding the premise of the current study, it is rather disorganized and is presented as a series of facts, including a few one sentence paragraphs, etc.  To make the study optimally appreciated by a broad readership, the introduction should be re-written and presented as a clear story.

3.       Figure 1 (and elsewhere):  Rather than stating percentage of proteins identified, it would be more accurate to state percentage of open reading frames (ORFs) since proteins per se are not identified by sequencing.  

see comments above

Author Response

Dear Reviewer

We like to thank you for the review of our manuscript ijms-2497987

Please notice our answers to the reviewer comments in the following point-by-point reply.

Reviewer

Response

1.        The manuscript should be very carefully reviewed for optimal use of the English language.

We fully agree with the reviewer and corrected the English language.

2.       The introduction, while it contains a great deal of important information regarding the premise of the current study, it is rather disorganized and is presented as a series of facts, including a few one sentence paragraphs, etc.  To make the study optimally appreciated by a broad readership, the introduction should be re-written and presented as a clear story.

We fully agree with the reviewer and are grateful for this comment. The introduction text has now been revised and we hope it looks better now.

3.       Figure 1 (and elsewhere):  Rather than stating percentage of proteins identified, it would be more accurate to state percentage of open reading frames (ORFs) since proteins per se are not identified by sequencing.

We fully agree with the reviewer and changed "proteins" to "ORFs", in several places in the text we changed "proteins" to "amino acid sequences".

Reviewer 3 Report

Manuscript ID: ijms-2497987

Review Report

This is a report for the manuscript entitled “RNA-seq virus fraction in Baikal and treated wastewaters”.

The manuscript is clear, well-written and provides comprehensive data concerning a relevant subject of general and human health interest.

Only few minor amendments are suggested in the PDF of the revised manuscript.

Author Response

Dear Reviewer

We like to thank you for the review of our manuscript ijms-2497987

Reviewer

Response

Please reformulate. Two is not the highest number of genome molecules. A protein shell (capsid) does not exist for many viruses.

We fully agree with the reviewer and we have reformulated the sentence.

"234 viruses known" ???

Dear reviewer we apologize for this incorrect English translation. We have corrected this sentence.

"234 known viruses"

OR: "bacterial nucleic acid"  ?

We fully agree with the reviewer and have corrected text.

Reviewer 4 Report

The authors present an interesting study on the viromes of the pelagic zone of Lake Baikal (sampled at three locations) and two points of discharge of treated wastewater into the system. Overall, the study results support the conclusions drawn. As a general comment, I think the authors should try to improve the linkage between the viruses identified and their potential hosts. While accepting in some cases this may not be known, a point that is not made until late in the manuscript. 

 Line 2 Suggest revision “Analyses of the pelagic viromes of Lake Baikal and treated inflow wastewaters using RNA-seq”

Line 8-24 I would suggest the authors review the abstract. It is currently overladen with results. A good abstract should give, some background, what the problem/question that is being addressed, very briefly describe the approach, summarise the key results (not present all the results, state the key conclusions, and then describe the importance of the study. This provides the potential reader of the manuscript with a good feel for the overall study. The current abstract is entirely results, except for the first sentence.

Line 28 suggest revision “Microorganisms are the”

Line 28 suggest revision “the aquatic biological system”

I am not sure “objects” is the best term. Does this statement take into account other phyla, such as invertebrates? A suitable reference would be good to support this statement.

Line 33 suggest deletion of “one or two” – there are far more variants with respect to viral genome composition.

Line 77 suggest revision receiving

Line 82 suggest replacing outbursts with outbreaks. I would also suggest the authors clarify what host or hosts are involved in these outbreaks.

Line 138 suggest replacing confirmed

Line 195 suggest revision Comparing the results from RVP4, RVP5, RVP6 obtained from

Line 218 The authors refer to the relevant host for the identified viruses, I think this approach should be utilised throughout the manuscript or at least when a specific class of virus is mentioned for the first time. This would help the reader to discern what might be consider a native virus (expected to be there) compared to a non-native virus (perhaps there due to contamination).

Line 227 How could the results for some samples be 100% for the Riboviria?

Line 185 Does this mean that none of the viruses detected have been reported in Baikal previously?

Line 294 Suggest adding a taxonomic group for this RdPp sequence.

Line 302 Suggest placing panel B below panel A. The legend could also be expanded to give a more detailed explanation of what is shown.

Line 390 suggest revision “Proteins with similarity to proteins of viruses known to infect animals”

Line 518-520 please review these sentences, I am not sure what the intended message is.

Line 574 suggest revision “capsid”

Line 668 suggest revision “despite the inadequate”

Line 677 suggest revision “effluent wastewater discharge points”

Line 679 suggest revision “viruses known to infect humans”

Overall the manuscript could do with a thorough read-through to ensure the intended messages are clear. I have highlighted some of these in my comments to the authors. I have not taken this into account in my decision regarding the manuscript.

Author Response

Dear Reviewer

We like to thank you for the review of our manuscript ijms-2497987

Please notice our answers to the reviewer comments in the following point-by-point reply.

Reviewer

Response

The authors present an interesting study on the viromes of the pelagic zone of Lake Baikal (sampled at three locations) and two points of discharge of treated wastewater into the system. Overall, the study results support the conclusions drawn. As a general comment, I think the authors should try to improve the linkage between the viruses identified and their potential hosts. While accepting in some cases this may not be known, a point that is not made until late in the manuscript.

We are grateful to the reviewer for his interest in our work. It is true that in many cases the detection of host range is a difficult task, also due to the limited number of cultured viruses.

Line 2 Suggest revision “Analyses of the pelagic viromes of Lake Baikal and treated inflow wastewaters using RNA-seq”

We thank the reviewer for the suggestion. We tried to avoid the words "analysis" and "characterization" in the title. But we have inserted the word "Lake" in the title of the manuscript.

Line 8-24 I would suggest the authors review the abstract. It is currently overladen with results. A good abstract should give, some background, what the problem/question that is being addressed, very briefly describe the approach, summarise the key results (not present all the results, state the key conclusions, and then describe the importance of the study. This provides the potential reader of the manuscript with a good feel for the overall study. The current abstract is entirely results, except for the first sentence.

We fully agree with the reviewer and have rewritten the abstract.

Line 28 suggest revision “Microorganisms are the”

We fully agree with the reviewer and have rewritten this sentence.

Line 28 suggest revision “the aquatic biological system”

We fully agree with the reviewer and have rewritten this sentence.

I am not sure “objects” is the best term. Does this statement take into account other phyla, such as invertebrates? A suitable reference would be good to support this statement.

We fully agree with the reviewer and have rewritten this sentence.

Line 33 suggest deletion of “one or two” – there are far more variants with respect to viral genome composition.

We fully agree with the reviewer and have corrected text.

Line 77 suggest revision “receiving”

We fully agree with the reviewer and replaced by "bringing".

Line 82 suggest replacing “outbursts” with “outbreaks”. I would also suggest the authors clarify what host or hosts are involved in these outbreaks.

We fully agree with the reviewer and have corrected text. The data presented in the original article implies analysis of human disease outbreaks.

Line 138 suggest replacing “confirmed”

We fully agree with the reviewer and have corrected text.

Line 195 suggest revision “Comparing the results from RVP4, RVP5, RVP6 obtained from”

We fully agree with the reviewer and have corrected text.

Line 218 The authors refer to the relevant host for the identified viruses, I think this approach should be utilised throughout the manuscript or at least when a specific class of virus is mentioned for the first time. This would help the reader to discern what might be consider a native virus (expected to be there) compared to a non-native virus (perhaps there due to contamination).

We thank the reviewer for this suggestion and have included the hosts.

Line 227 How could the results for some samples be 100% for the Riboviria?

This chapter focuses on the RNA-contained viruses identified using the NR database and among all sequences detected in the BVP6 sample, all sequences were assigned to the realm of Riboviria and had no higher taxonomic rank.

Line 185 Does this mean that none of the viruses detected have been reported in Baikal previously?

This is probably true, since this database (IMG/VR) in addition uses metagenomic assemblies, which are probably absent in the NR database. While we used this database for the first time in our work on the Baikal sequences.

Line 294 Suggest adding a taxonomic group for this RdPp sequence.

The taxonomic annotation of all identified RdRp is summarized in the TableS2.

Line 302 Suggest placing panel B below panel A. The legend could also be expanded to give a more detailed explanation of what is shown.

We thank the reviewer for this suggestion and we reorganized the figure and expanded the description in the legend. We hope that it is now clearer.

Line 390 suggest revision “Proteins with similarity to proteins of viruses known to infect animals”

We fully agree with the reviewer and have corrected text. In addition, we have corrected "proteins" to ORFs as per the reviewer's 2 comment.

Line 518-520 please review these sentences, I am not sure what the intended message is.

We fully agree with the reviewer and have removed these sentences.

Line 574 suggest revision “capsid”

We fully agree with the reviewer and have corrected text.

Line 668 suggest revision “despite the inadequate”

We fully agree with the reviewer and have corrected text.

Line 677 suggest revision “effluent wastewater discharge points”

We fully agree with the reviewer and have corrected text.

Line 679 suggest revision “viruses known to infect humans”

We fully agree with the reviewer and have corrected text.

Overall the manuscript could do with a thorough read-through to ensure the intended messages are clear. I have highlighted some of these in my comments to the authors. I have not taken this into account in my decision regarding the manuscript.

We thank the reviewer for their interest in the work, and hope that the corrected version will be fully compliant with all requirements.

Reviewer 5 Report

The work presented for review is very interesting. It describes little known to date analysis of the viriom of the aquatic environment. The presented bioinformatics analysis is carried out in a very professional and detailed manner.

The main issue analyzed in the presented manuscript concerns the analysis of viruses present in the aquatic environment, along with the analysis of possible pathogenic entities. This topic is very interesting and fills in the gaps in the knowledge we have about the microbiome of this environment. This is because research to date has mainly focused on the analysis of the microbiome (bacteria, fungi). The data obtained adds to our knowledge on the subject. In addition, the juxtaposition of data coming from different environments allows for comparison. In the future, I suggest trying to isolate the viral particles themselves to rid the DNA material of eukaryotic organisms. Sequencing material from viruses and microorganisms at the same time could significantly affect the amount of data we obtain for the former group. The conclusions section includes all the necessary information. The references cited are appropriate and in line with the topic. The quality of the figures and charts is acceptable.

Author Response

Dear Reviewer

We like to thank you for the review of our manuscript ijms-2497987

Reviewer

Response

The work presented for review is very interesting. It describes little known to date analysis of the viriom of the aquatic environment. The presented bioinformatics analysis is carried out in a very professional and detailed manner.

The main issue analyzed in the presented manuscript concerns the analysis of viruses present in the aquatic environment, along with the analysis of possible pathogenic entities. This topic is very interesting and fills in the gaps in the knowledge we have about the microbiome of this environment. This is because research to date has mainly focused on the analysis of the microbiome (bacteria, fungi). The data obtained adds to our knowledge on the subject. In addition, the juxtaposition of data coming from different environments allows for comparison. In the future, I suggest trying to isolate the viral particles themselves to rid the DNA material of eukaryotic organisms. Sequencing material from viruses and microorganisms at the same time could significantly affect the amount of data we obtain for the former group. The conclusions section includes all the necessary information. The references cited are appropriate and in line with the topic. The quality of the figures and charts is acceptable.

We thank the reviewer for their interest in our work and for their appreciation. Indeed, it is currently very difficult to separate viruses from other components for better analysis, but in general we hope that with the development of technology and new reagents it will be possible in the future. At present we have tried to analyze viral sequences on a modern level and we hope that our work will help many scientists working in this field.